# A small-molecule ICMT inhibitor delays senescence of Hutchinson-Gilford progeria syndrome cells

Xue Chen[1,2], Haidong Yao[1], Muhammad Kashif[1], Gwladys Revêchon[1], Maria Eriksson[1], Jianjiang Hu[1], Ting Wang[1], Yiran Liu[1], Elin Tüksammel[1], Staffan Strömblad[1], Ian M Ahearn[3], Mark R Philips[4], Clotilde Wiel[1], Mohamed X Ibrahim[1,5]*, Martin O Bergo[1]*

[1]Department of Biosciences and Nutrition, Karolinska Institutet, Huddinge, Sweden; [2]Department of Plastic and Cosmetic Surgery, Tongji Hospital, Tongji Medical College, Huazhong University of Science and Technology, Wuhan, China; [3]Department of Dermatology, New York University Grossman School of Medicine, New York, United States; [4]Perlmutter Cancer Center, New York University Grossman School of Medicine, New York, United States; [5]Sahlgrenska Center for Cancer Research, Gothenburg, Sweden

**Abstract** A farnesylated and methylated form of prelamin A called progerin causes Hutchinson-Gilford progeria syndrome (HGPS). Inhibiting progerin methylation by inactivating the isoprenylcysteine carboxylmethyltransferase (ICMT) gene stimulates proliferation of HGPS cells and improves survival of *Zmpste24*-deficient mice. However, we don't know whether *Icmt* inactivation improves phenotypes in an authentic HGPS mouse model. Moreover, it is unknown whether pharmacologic targeting of ICMT would be tolerated by cells and produce similar cellular effects as genetic inactivation. Here, we show that knockout of *Icmt* improves survival of HGPS mice and restores vascular smooth muscle cell numbers in the aorta. We also synthesized a potent ICMT inhibitor called C75 and found that it delays senescence and stimulates proliferation of late-passage HGPS cells and *Zmpste24*-deficient mouse fibroblasts. Importantly, C75 did not influence proliferation of wild-type human cells or *Zmpste24*-deficient mouse cells lacking *Icmt*, indicating drug specificity. These results raise hopes that ICMT inhibitors could be useful for treating children with HGPS.

*For correspondence:
mohamed.ibrahim@gu.se (MXI);
martin.bergo@ki.se (MOB)

**Competing interests:** The authors declare that no competing interests exist.

## Introduction

Hutchinson-Gilford progeria syndrome (HGPS) is caused by the accumulation of progerin, a mutant form of prelamin A that is farnesylated and methylated within the nuclear envelope (*De Sandre-Giovannoli et al., 2003*; *Eriksson et al., 2003*). Farnesyltransferase inhibitors (FTIs) prevent progerin farnesylation and improve some clinical phenotypes of HGPS patients, including survival, but the effect is modest (*Gordon et al., 2018*; *Young et al., 2005*). Also, a potential limitation of this approach is that FTIs are anti-proliferative (*Lee et al., 2010*), and children with progeria would benefit from a therapy that supports cell proliferation. We found earlier that inhibiting the methylation of progerin by inactivating the isoprenylcysteine carboxylmethyltransferase (ICMT) gene overcomes senescence and increases proliferation of HGPS cells (*Ibrahim et al., 2013*). Also, a knockout of *Icmt* substantially improves clinical phenotypes and survival of *Zmpste24*-deficient mice, a model of progeria (*Ibrahim et al., 2013*). This result raises the possibility that inhibiting ICMT activity could be a useful therapeutic strategy. An important step in the preclinical validation of this strategy would be to

determine whether knockout of *Icmt* improves phenotypes and survival in an authentic progerin-expressing HGPS mouse model. Another step would be to determine whether pharmacologic targeting of ICMT produces similar cellular effects as genetic inactivation. To address these issues, we defined the consequences of knocking out *Icmt* in progerin-knock-in mice; and we synthesized a potent cell-permeable ICMT inhibitor, compound 75 (C75) (*Judd et al., 2011*), and examined its effects on HGPS cells.

## Results

We bred mice with a hypomorphic *Icmt* allele (*Icmt*hm; with ~85% reduced ICMT activity [*Ibrahim et al., 2013*]) with progerin-expressing lamin A knock-in mice (*Lmna*G609G [*Osorio et al., 2011*]). As expected from previous studies, *Lmna*G609G/G609G*Icmt*+/+ mice developed alopecia, stunted growth, and weight loss, and all mice had died by 129 days of age; at that time, numbers of vascular smooth muscle cell (VSMC) nuclei in aortic arch sections were reduced by ~75% and muscle fiber size in the quadriceps muscle were 50% smaller compared with wild-type mice (*Figure 1a–f*). In contrast, at 129 days of age, the *Lmna*G609G/G609G*Icmt*hm/hm mice were still alive with substantially higher body weights, and when they were sacrificed, VSMC numbers were found to be normalized and skeletal muscle fiber size increased (*Figure 1a–f*). Although these data are statistically sound,

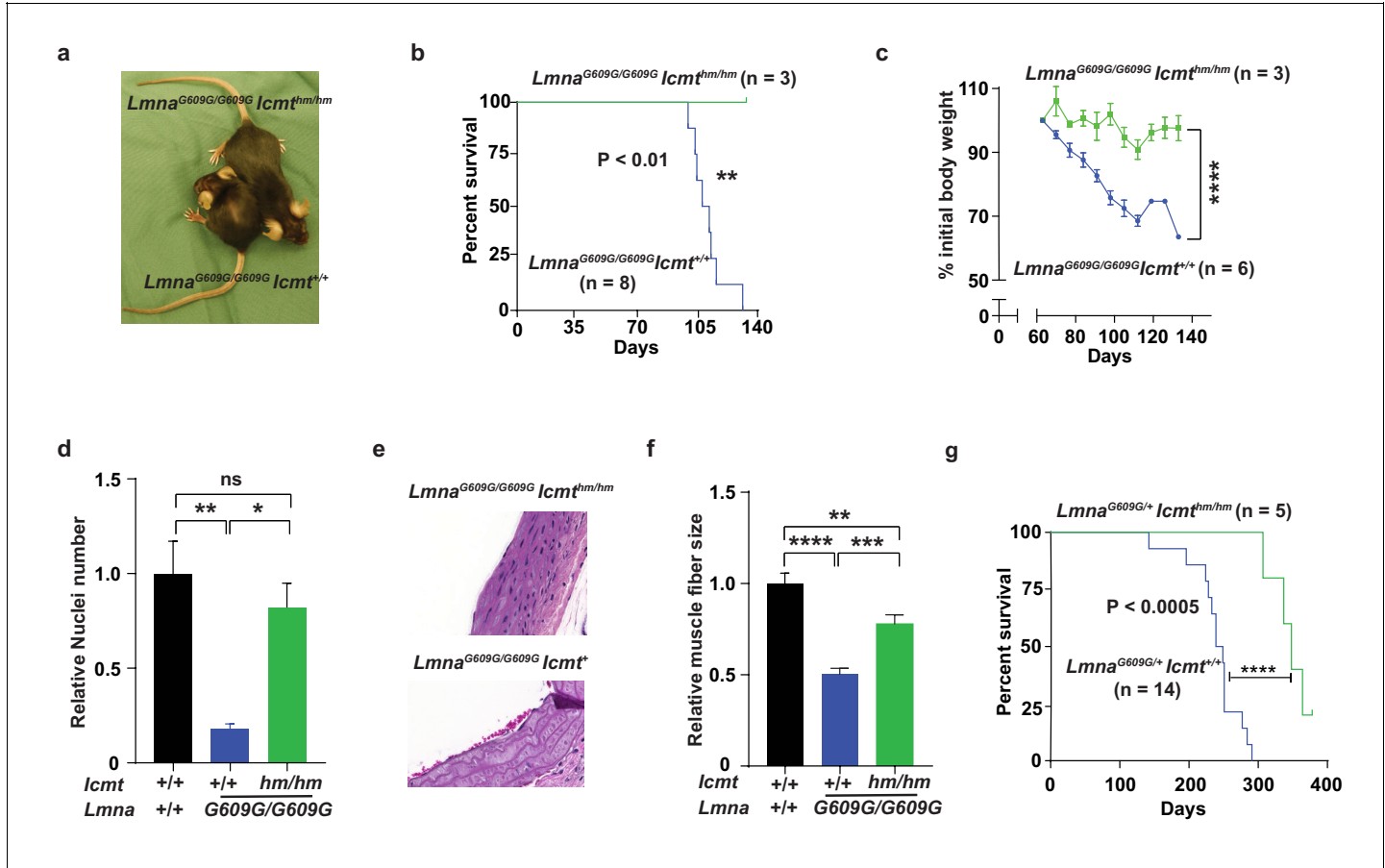

**Figure 1.** Targeting *Icmt* improves survival and aorta and muscle phenotypes of progerin-knock-in mice. (a) Photograph of 15-week-old littermate female mice. (b) Kaplan-Meier plot showing survival of *Lmna*G609G/G609G*Icmt*hm/hm (n = 3) and *Lmna*G609G/G609G*Icmt*hm/+ (n = 8) mice; the three double-homozygotes were killed for analyses when all the control mice had died of progeria. (c) Body weight curves of mice in panel B. (d) Number of vascular smooth muscle cell nuclei in medial layer of aortic arch sections. Data are mean of three mice/genotype. (e) Representative photographs of aortic arch sections from panel d. (f) Skeletal muscle fiber cross-sectional diameter. Data are mean of 50 independent muscle cells diameters in *M. quadriceps extensor* from three mice/genotype. (g) Kaplan-Meier plot showing survival of *Lmna*G609G/+*Icmt*hm/hm (n = 5) and *Lmna*G609G/+*Icmt*+/+ (n = 14) mice. *p<0.05; **p<0.01; ***p<0.005; ****p<0.001; n.s., not significant.

they should be interpreted with caution as the mice were difficult to breed and we only obtained three double homozygotes. Consequently, we also analyzed $Lmna^{G609G/+}Icmt^{+/+}$ mice, which had a maximal life span of 290 days (*Figure 1g*) (and no aorta and skeletal muscle phenotypes); importantly, all $Lmna^{G609G/+}Icmt^{hm/hm}$ mice were still alive at 290 days and their overall survival were increased (*Figure 1g*). These results are important because progerin rather than prelamin A causes progeria in $Lmna^{G609G}$ mice, and because homozygous $Lmna^{G609G/G609G}$ mice exhibit a vascular phenotype, which are prominent in children with HGPS, but absent in *Zmpste24*-deficient mice used in earlier studies (*Ibrahim et al., 2013*)—and *Icmt* inactivation markedly improved this phenotype.

We next synthesized the ICMT inhibitor C75 as described (*Judd et al., 2011*), and found that its IC50 was 0.5 µM (*Figure 2a—figure 2 supplement S1*). Prolonged C75 incubation was well tolerated by two different human HGPS cell lines and caused prelamin A accumulation and mislocalization of the RAS oncogene —markers of reduced ICMT activity—but did not affect the nuclear shape abnormalities (*Figure 2b–d—figure 2 supplement S2a*). C75 did not influence the electrophoretic mobility of HDJ2 which could have been indicative of effects on FTase-mediated farnesylation (*figure 2 supplement S2b*). Importantly, C75 increased proliferation of late-passage HGPS cell lines as judged by 45- to 70-day population-doubling assays (*Figure 2e–f*). The drug had increased cell viability already at 8 days, that is, before the increase in cell proliferation was evident (*figure 2 supplement S2c–d*). C75 also increased proliferation of *Zmpste24*-deficient mouse fibroblasts with normal *Icmt* expression but not in cells lacking *Icmt*, indicating drug specificity (*Figure 2g–h*). The drug did not affect proliferation of wild-type human fibroblasts (*Figure 2i*). In contrast, the FTI lonafarnib rapidly reduced proliferation of HGPS cells and abolished the effect of C75 on cell growth (*Figure 2j–k*). The latter finding makes sense because protein methylation cannot occur without protein farnesylation.

Consistent with increased proliferation, C75 increased the fraction of HGPS cells in the G1 and S/G2/M phases of the cell cycle (*figure 2 supplement S2e*); reduced senescence-associated β-galactosidase activity (*Figure 2l–m*); and normalized the expression of the senescence markers *IL6* and *CDKN2A* (*Figure 2o–p*). The drug also normalized oxygen consumption rates and ATP production in HGPS cells, as judged by Seahorse analyses; and reduced the levels of oxidative stress (*figure 2 supplement S2f–g*). In contrast, C75 did not influence expression of endoplasmic reticulum (ER) stress markers in HGPS cells; and DNA damage signaling markers, including γ-$H_2$AX, remained unchanged (*figure 2 supplement S2h–i*). C75 did however reduce the fraction of cells with nuclei harboring multiple γ-$H_2$AX foci, examined by immunofluorescence, but not the total fraction of γ-$H_2$AX-positive nuclei (*figure 2 supplement S2j*). A potential interpretation of the latter finding is that C75 increases proliferation of cells with low levels of DNA damage which outcompete cells with high levels.

The signaling molecule AKT binds progerin and farnesyl-prelamin A (in HGPS and *Zmpste24*-deficient cells, respectively) and exhibits only low levels of phosphorylation (*Ibrahim et al., 2013*). C75 reduced the progerin–AKT interactions and increased AKT phosphorylation (*Figure 3a–d*), but C75 did not influence phospho-AKT levels in wild-type cells (*Figure 3b*). Although these data don't reveal whether AKT is functionally involved in the improved phenotypes upon C75 administration, we showed previously that pharmacologic AKT inhibition prevents the increased proliferation following *Icmt* knockout in mouse cells (*Ibrahim et al., 2013*). Despite improvements in multiple cellular phenotypes, C75 increased the absolute levels of progerin in HGPS cells, a consequence of reduced progerin turnover (*Figure 3e–g*). Moreover, C75 mislocalized some progerin and farnesyl-prelamin A away from the nuclear membrane into the nucleoplasm (*Figure 3h–i*).

## Discussion

We conclude that genetic *Icmt* inactivation improves survival and unique phenotypes of an authentic HGPS mouse model. We further conclude that pharmacologic inhibition of ICMT delays senescence, restores respiration rates and ATP production, and stimulates proliferation of HGPS and *Zmpste24*-deficient cells, most of which is consistent with findings in *Icmt*-deficient cells (*Ibrahim et al., 2013*). Blocking methylation partially mislocalizes progerin to the nucleoplasm, disrupts its interaction with AKT, and increases AKT signaling. These positive phenotypes of blocking methylation with C75 outweighed any potential adverse effects from the modest amount of progerin accumulation. In wild-type cells, prelamin A is fully processed to mature lamin A and the cells grow and proliferate normally. In HGPS and *Zmpste24*-deficient cells, farnesylated and methylated progerin/prelamin A

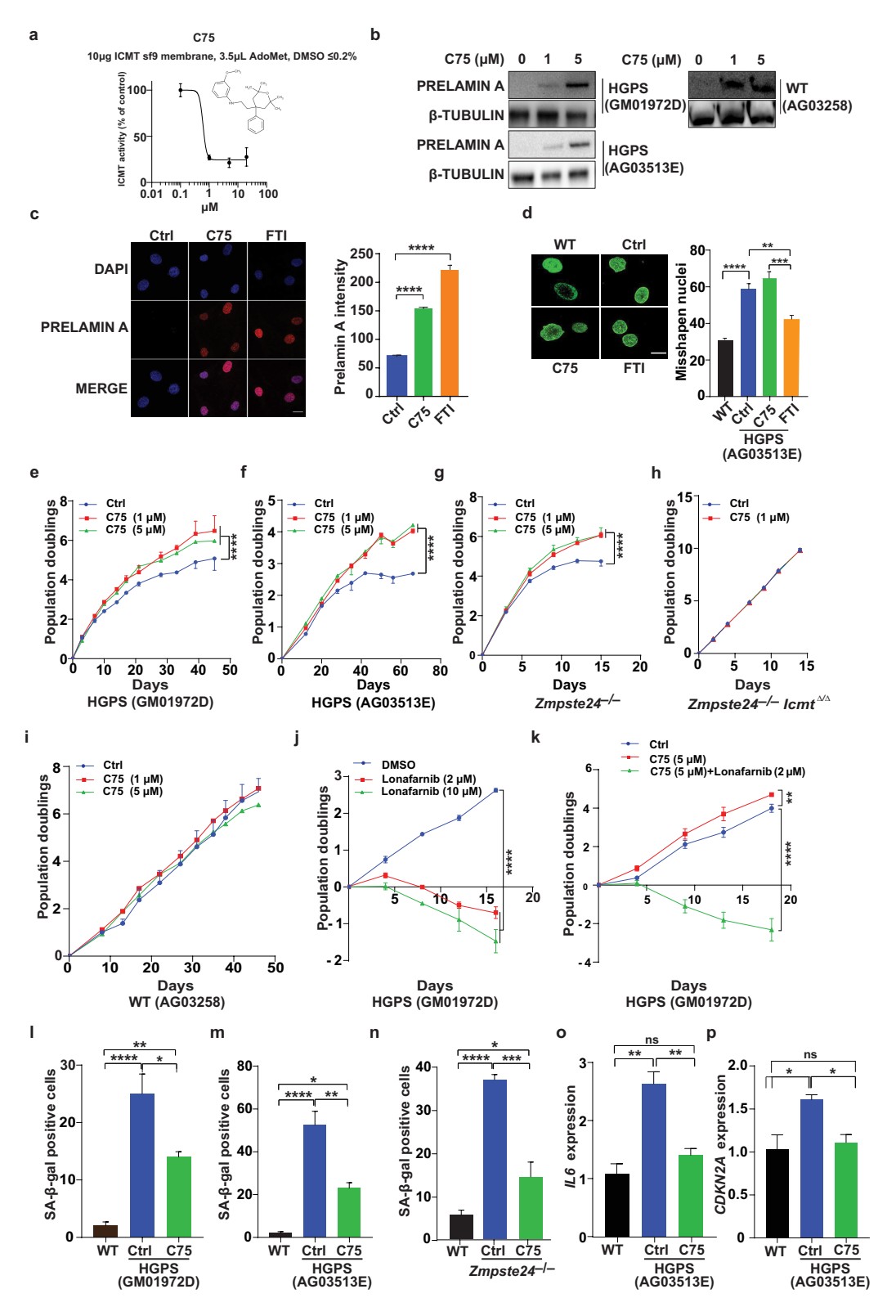

**Figure 2.** C75 inhibits isoprenylcysteine carboxylmethyltransferase (ICMT) activity and increases proliferation and reduces senescence of HGPS cells without affecting nuclear shape. (a) Chemical structure of compound 75 (C75) and percent *ICMT* activity remaining after incubation with C75. (b) Western blot showing increased amounts of prelamin A in Hutchinson-Gilford progeria syndrome (HGPS) and wild-type (WT) cells incubated with C75 for 20 days; β tubulin was the loading control. (c) Left, immunofluorescence images showing prelamin A expression in HGPS cells incubated with 5 µM

*Figure 2 continued on next page*

*Figure 2 continued*

C75 for 20 days and 10 µM farnesyltransferase inhibitor (FTI) for 3 days; the cells were counterstained with 4′,6-diamidino-2-phenylindole (DAPI). Right, quantification of prelamin A staining intensity (n = 2 cell lines and six individual images/cell line). (d) Left, representative nuclei of LAP2β-stained WT cells and HGPS cells incubated with vehicle (Ctrl), 5 µM C75 for 20 days, and 10 µM FTI for 3 days. Right, quantification of misshapen nuclei. Data are mean of ~1000 nuclei and two independent experiments per cell line and condition. (e–h) Population doubling assays of late-passage HGPS cell lines (e, f), primary *Zmpste24*–/– mouse fibroblasts (g), and *Zmpste24*–/–*Icmt*–/– fibroblasts (h) incubated with vehicle (Ctrl) and C75. (i) Population doubling of WT cells incubated with C75. (j, k) Population doubling assays of a late-passage HGPS cell line incubated with vehicle (Ctrl), lonafarnib, C75, and both drugs. (l–n) Senescence-associated beta-galactosidase (β-gal) staining of WT, HGPS, and *Zmpste24*–/– fibroblasts incubated with vehicle (Ctrl) and C75 for 20 days. (o, p) Interleukin 6 (*IL-6*) (o) and *CDKN2A* (p) expression in cells from experiment in panel *m*. \*\*p<0.01; \*\*\*p<0.001; \*\*\*\*p<0.0001; n.s., not significant.

The online version of this article includes the following figure supplement(s) for figure 2:

**Figure supplement 1.** C75 synthesis and absorption, distribution, metabolism, and excretion (ADME) test.

**Figure supplement 2.** C75 increases viability, proliferation, and metabolic rates of Hutchinson-Gilford progeria syndrome (HGPS) cells and reduces reactive oxygen species (ROS) levels and nuclei with high levels of DNA damage.

accumulates and causes senescence. Our data suggest that progerin/prelamin A methylation contributes to the toxicity of these proteins and their ability to induce senescence, and we propose that blocking progerin/prelamin A methylation mislocalizes the proteins into the nucleoplasm and thereby reduces their ability to induce DNA damage, metabolic alterations, and senescence.

A limitation of C75 is that despite good apparent permeability (*Figure 2 — supplement 1d*) it is predicted to have poor bioavailability (i.e., very hydrophobic and high first-passage metabolism in in silico ADME analyses). Thus, new compounds will be required for in vivo studies in mice. Nonetheless, our study takes two important steps in the preclinical validation of ICMT as a potential drug target, and thereby raises hopes that ICMT inhibition could be an effective strategy for treating children with HGPS and progeroid disorders resulting from *ZMPSTE24* deficiency (*Michaelis and Hrycyna, 2013*).

## Materials and methods

### Mice

*Icmt*$^{hm/hm}$ mice (*Bergo et al., 2004*; *Wahlstrom et al., 2008*) were bred with *Lmna*$^{G609G/G609G}$ mice (*Osorio et al., 2011*) to produce *Lmna*$^{G609G/G609G}$*Icmt*$^{hm/hm}$ mice. Controls were littermate *Lmna*$^{G609G/G609G}$*Icmt*$^{hm/+}$ mice and *Lmna*$^{G609G/G609G}$*Icmt*$^{+/+}$ mice, which were indistinguishable in phenotype and collectively designated *Lmna*$^{G609G/G609G}$*Icmt*$^{+/+}$. We also used *Lmna*$^{G609G/+}$*Icmt*$^{hm/hm}$ mice and the controls *Lmna*$^{G609G/+}$*Icmt*$^{hm/+}$ and *Lmna*$^{G609G/+}$*Icmt*$^{+/+}$ mice, which were collectively designated *Lmna*$^{G609G/+}$*Icmt*$^{+/+}$. Genotyping was performed by polymerase chain reaction (PCR) on genomic DNA from ear or tail biopsies. Mice were monitored daily and weighed weekly. The aortic arch and quadriceps muscle were harvested and fixed in 10% formalin for 24 hr and then stained with hematoxylin and eosin. VSMC nuclei in the aortic media and skeletal muscle fiber size were quantified as described (*Greising et al., 2012*; *Kim et al., 2018*). Mouse experiments were approved by the Animal Research Ethics Committees in Gothenburg and Linköping, Sweden.

### Drug synthesis, ICMT activity assay, and apparent permeability assay

C75 was synthesized by Recipharm AB as described (*Judd et al., 2011*). ICMT activity was carried out as described (*Choy and Philips, 2000*; *Zhou et al., 2016*). Apparent permeability assay was carried out by analyzing the apical-to-basolateral (and vice versa) transport of C75 using Caco-2 cell monolayers as described (*Hubatsch et al., 2007*).

### Cell culture and cell proliferation

Human HGPS cell lines (GM01972D and AG03513E) and control cell lines from unaffected parents (AG03258 and AG03512) were from the Coriell Institute. Primary mouse fibroblasts (MEFs) isolated from E13.5–E14.5 *Zmpste24*-deficient embryos (*Bergo et al., 2002*) were cultured in low-glucose Dulbecco's modified eagle medium DMEM (21885025, ThermoFisher) supplemented with 10% fetal bovine serum (26140079, ThermoFisher), 1% penicillin/streptomycin (15070063, ThermoFisher), and

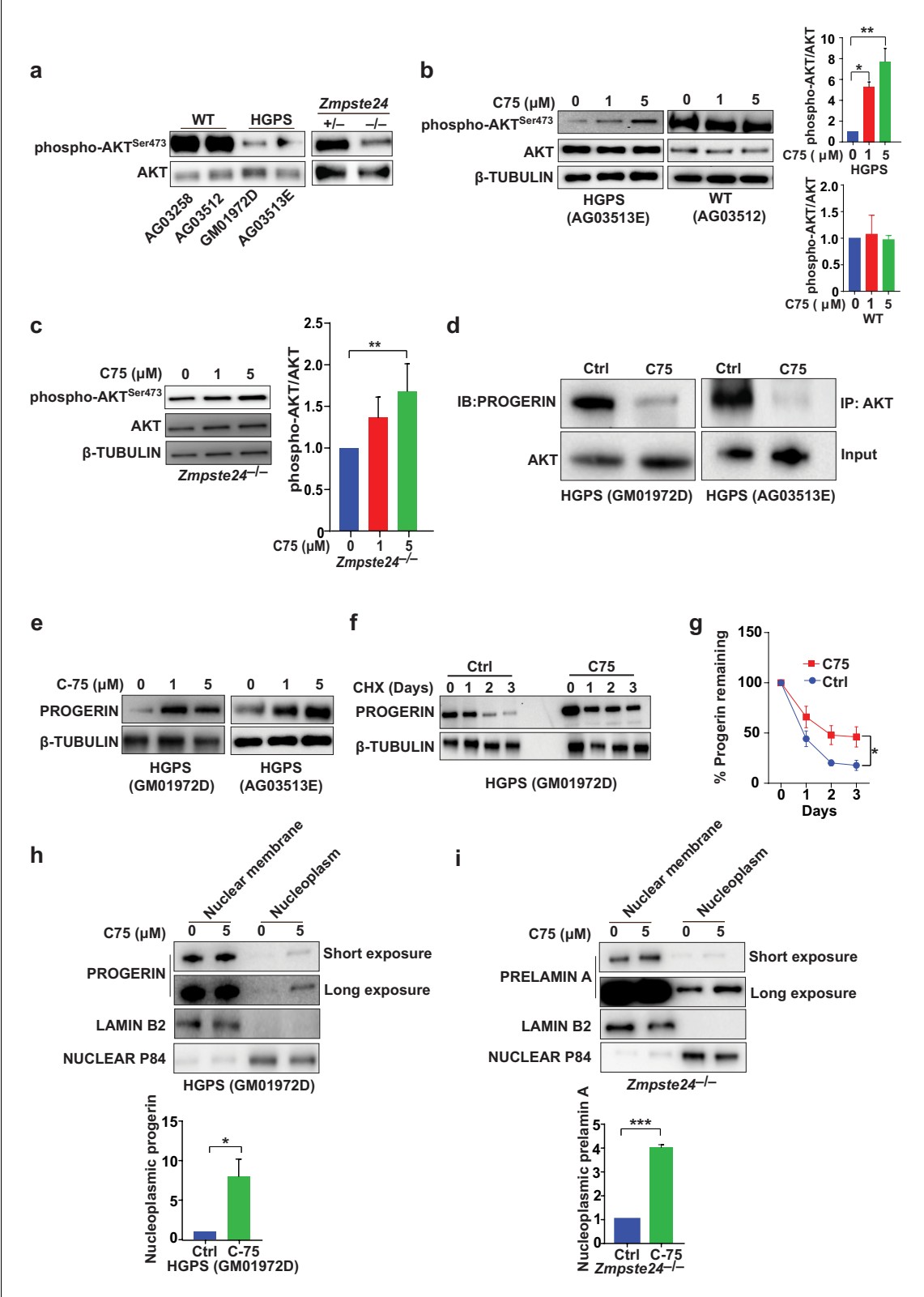

**Figure 3.** The isoprenylcysteine carboxylmethyltransferase inhibitor C75 disrupts progerin-AKT interactions, increases AKT activation, and mislocalizes progerin from the nuclear membrane to the nucleoplasm. (a) Western blot showing amounts of phosphorylated and total AKT (a.k.a protein kinase B) in whole-cell lysates of human wild-type (WT) and Hutchinson-Gilford progeria syndrome (HGPS) cells and mouse *Zmpste24*-deficient fibroblasts. (b, c) Left panels, western blots (WB) showing amounts of phosphorylated and total AKT in WT and HGPS cells (b) and *Zmpste24*-deficient (c) fibroblasts

*Figure 3 continued on next page*

**Figure 3 continued**

incubated with C75 for 20–30 days; β tubulin was the loading control. Right panels, quantification of the ratio of phosphorylated and total AKT from densitometry analyses of protein bands. Data are mean of two independent cell lines, each analyzed in duplicate. (d) Immunoprecipitation (IP) and WB analyses showing that C75 disrupts the association between AKT and progerin. The lysates were also used directly for WB with total AKT antibodies (input). (e) WB showing amounts of progerin in HGPS cells incubated with C75 for 20 days. (f) WB showing amounts of progerin remaining in HGPS cells incubated with vehicle and 5 μM C75 for 20 days and then with cycloheximide to stop protein synthesis. (g) Quantification of progerin amounts from the experiment in panel f and two others like it. (h, i) Left panels, WB showing amounts of progerin and prelamin A in nuclear membrane and nucleoplasm fractions of HGPS (h) and *Zmpste24*-deficient (i) fibroblasts, respectively, incubated with vehicle and 5 μM C75 for 20 days. Lamin B2 and nuclear P84 were loading controls for nuclear membrane and nucleoplasm fractions, respectively. Right panels, quantification of nucleoplasmic progerin/prelamin A from densitometry analyses. *p<0.05; **p<0.01; ***p<0.001.

1% MEM non-essential amino acid (11140068, ThermoFisher). Cell viability proliferation assays were carried out by plating $1 \times 10^3$ cells in 96-well plates. Cell viability was determined every 3 days with PrestoBlue Cell Viability Reagent (A13262, ThermoFisher); absorbance at 570 and 600 nm was measured with the Multi-mode reader (BioTek). Population doubling assays were carried out by seeding $3 \times 10^5$ cells on 10 cm plates. The cells were trypsinized, counted, and re-seeded every 3 days (MEFs) or 4–8 days (human cells). Lonafarnib (2 and 10 μM, SML1457; Sigma-Aldrich) was used in some population doubling assays. All cell lines tested negative for mycoplasma.

## Immunofluorescence and nuclear shape

Wild-type and HGPS cell lines were cultured in Glass Bottom Microwell Dishes (MatTek) for 24 hr, fixed in 4% paraformaldehyde, permeabilized with 0.4% Triton X-100, and blocked with PBS containing 2% bovine serum albumin. The cells were incubated overnight with antibodies to LAP2β (1:100, 611000, BD Biosciences), Prelamin A (1:400, MABT345, Millipore), and phospho-γH2AX (1:800, 05–636, Millipore), followed by incubation for 1 hr with secondary antibodies (1:1000, Alexa Fluor Plus 488 goat anti-mouse IgG, SA243833; 1:1000, Alexa Fluor 594 donkey anti-rat IgG, 1870948, Life Technologies) and staining with 4′,6-diamidino-2-phenylindole (DAPI) (1:500, 62248, ThermoFisher) for 15 min. Prelamin A staining intensity and the frequency of misshapen nuclei were quantified as described (*Ibrahim et al., 2013*). γH2AX foci were counted manually. FTI-276 (2 μM, F9553; Sigma-Aldrich) was used as a positive control for prelamin A accumulation and for restoring nuclear shape abnormalities.

## Immunoblots and immunoprecipitation

Cells were lysed in buffer containing 9 M urea (U0631, Sigma-Aldrich) and complete protease inhibitor cocktail (78430, ThermoFisher), sonicated, and cleared by centrifugation (14,000 × *g* for 10 min). The lysates were size-fractionated on 10% Mini-PROTEAN TGX Stain-Free gels (456–8036, Bio-Rad) and transferred to nitrocellulose membranes (0.2 μm, 1704158, Bio-Rad). The membranes were incubated with primary antibodies overnight at 4°C and with secondary antibodies for 1 hr at room temperature. Primary antibodies were prelamin A (1:1000, MABT345, Millipore), progerin (1:500, 05–1231, Millipore), phospho-AKTSer473 (1:1000, 4060, Cell Signaling), total AKT (1:1000, 9272, Cell Signaling), pan-RAS (1:1000, ab69747, Abcam), lamin B2 (1:500, 33–2100, ThermoFisher), nuclear matrix protein p84 (1:1000, GTX70220, GeneTex), p21 (1:1000, sc-397, Santa Cruz Biotechnology), p53 (1:1000, sc-126, Santa Cruz Biotechnology), phospho-γH2AX (1:500, 05–636, Millipore), phospho-p53Ser15 (1:1000, 700439, ThermoFisher), antibodies in the ER stress sample kit (9956T, Cell Signaling), and β-tubulin (1:1000, T2200, Sigma-Aldrich). Secondary antibodies were anti-mouse (1:6000, 115-035-003, Jackson ImmunoResearch), anti-rabbit (1:6000, 111-035-003, Jackson ImmunoResearch), and anti-rat (1:6000, A9542, Sigma-Aldrich). Protein bands were detected and quantified on a ChemiDoc Touch Imaging System with Image lab (version 5.2.1). Cytosolic and membrane fractions were isolated with Qproteome Cell Compartment Kit (37502, QIAGEN). Immunoprecipitation (IP) was performed with the Dynabeads Protein G IP kit (100-07D, Life Technologies). To quantify progerin turnover rate, cells were incubated with cycloheximide (20 μg/ml, C4859, Sigma-Aldrich) to stop protein synthesis; lysates were prepared as above.

## Senescence-associated β-galactosidase assay

Senescence-associated β-galactosidase (SA-β Gal) staining on primary MEFs and human HGPS cell lines was performed using the Senescence Detection kit (9860, Cell Signaling). Cells were incubated with SA-β Gal solution for 24 hr (mouse) and 4 hr (human), separately, at 37°C. Results are reported as percent of blue cells.

## Quantitative PCR

RNA was isolated with the RNeasy Plus Mini kit (74136, QIAGEN) and cDNA was synthesized with the iScript cDNA synthesis kit (170–889, Bio-Rad). IL6 and CDKN2a expression was analyzed by reverse transcription quantitative PCR on a CFX384 Real-Time System (Bio-Rad) using Taqman human probe sets for IL6 (Hs00174131_m1, ThermoFisher) and CDKN2a (Hs01059210_m1, Thermo-Fisher). β-Tubulin (Hs00801390, ThermoFisher) was the reference gene.

## Isolation of nuclear membrane and nucleoplasm fractions

Nuclear membrane and nucleoplasm separation was performed on MEFs and human fibroblasts using Minute Nuclear Envelop Protein Extraction Kit (NE-013, Invent Biotechnology), and Minute Detergent-Free Nucleoplasm Isolation Kit (NI-024, Invent Biotechnology).

## Mitochondrial function assay

Mitochondrial function parameters were measured with the Cell Mito Stress Test kit using the Seahorse XFe96 Analyzer (Agilent). Cells were seeded in microplates (15,000 cells/well) (101085–004, Agilent) and cultured overnight at 37°C in a $CO_2$ incubator. Freshly prepared DMEM-base medium supplemented with glucose, pyruvate, and glutamine, and adjusted to pH 7.4 were added to the cells and they were incubated for 45 min at 37°C in a non-CO2 incubator and then analyzed at 37°C in the XFe96 Analyzer. Basal and maximal respiration and ATP production data were normalized to viable cell numbers obtained from identically treated additional wells using the Presto Blue Cell Viability assay (A13262, ThermoFisher).

## ROS measurements

HGPS cells were incubated with C75 for 20 days and then seeded in white 96-well plates ($5 \times 10^3$ cells/well). ROS measurements were performed using the H2DCFDA (H2-DCF, DCF) kit (D399, Thermofisher) in FluoroBrite medium (A18967-01, Life Technologies). Fluorescence (Excitation and Emission: 492–495/517–527) was recorded with a Synergy multimode reader (BioTek).

## Flow cytometry analysis

HGPS cells were incubated with 250 μl fixation/permeabilization solution (554714, BD) for 30 min on ice, in the dark; washed twice; incubated with antibodies to Ki67 (5 μl/sample; 561277, BD) at room temperature for 1 hr and 45 min; washed with PBS + 10% FCS; stained with 7AAD (5 μg/sample; A9400-1MG, Sigma) at room temperature for 20 min; resuspended and filtered into flow tubes; and analyzed using a BD LSRFortessa X-20.

## Statistics

Data are presented as mean ± SEM. For statistical analyses, we used Graphpad Prism software v.7; the log-rank test was used for survival, two-way ANOVA for cell-growth curves, one-way ANOVA with Bonferroni's post-hoc test when comparing three or more groups, and Student's $t$ test when comparing two groups only. Experiments were repeated 2–4 times unless stated otherwise; n indicates biological replicates.

## Acknowledgements

We thank Dr. C López-Otín for the *Lmna*[G609G] mice; Dr. X Xu for technical assistance; and Dr. S Young for helpful discussions. Microscopy was performed at the LCI facility/Nikon Center of Excellence, Karolinska Institutet, supported by grants from the Knut and Alice Wallenberg Foundation, Swedish Research Council, KI infrastructure, Centre for Innovative Medicine, and Jonasson Center at

the Royal Institute of Technology. The study was supported by grants from the Progeria Research Foundation, Center for Innovative Medicine (CIMED), and the Swedish Research Council (to MOB).

## Additional information

### Funding

| Funder | Author |
|---|---|
| Progeria Research Foundation | Martin O Bergo |
| Vetenskapsrådet | Martin O Bergo |
| Center for innovative medicine (CIMED), Karolinska Institutet, Huddinge, Sweden | Martin O Bergo |

The funders had no role in study design, data collection and interpretation, or the decision to submit the work for publication.

### Author contributions

Xue Chen, Data curation, Formal analysis, Investigation, Methodology, Writing - original draft, Project administration, Writing - review and editing; Haidong Yao, Data curation, Formal analysis, Supervision, Investigation, Writing - original draft, Project administration; Muhammad Kashif, Software, Formal analysis, Supervision, Writing - original draft, Project administration; Gwladys Revêchon, Ting Wang, Yiran Liu, Project administration; Maria Eriksson, Supervision, Writing - review and editing; Jianjiang Hu, Software, Project administration; Elin Tüksammel, Methodology, Project administration; Staffan Strömblad, Supervision, Methodology, Writing - review and editing; Ian M Ahearn, Investigation, Methodology; Mark R Philips, Supervision, Project administration, Writing - review and editing; Clotilde Wiel, Data curation, Formal analysis, Supervision, Methodology, Writing - original draft, Writing - review and editing; Mohamed X Ibrahim, Conceptualization, Data curation, Formal analysis, Supervision, Investigation, Methodology, Writing - original draft, Writing - review and editing; Martin O Bergo, Conceptualization, Data curation, Supervision, Funding acquisition, Methodology, Writing - original draft, Writing - review and editing

### Author ORCIDs

Staffan Strömblad (iD) https://orcid.org/0000-0002-1236-6339
Mohamed X Ibrahim (iD) https://orcid.org/0000-0002-7762-1580
Martin O Bergo (iD) https://orcid.org/0000-0002-6915-7140

### Ethics

Animal experimentation: Mouse experiments were in strict accordance with EU and Swedish law and were approved by the research animal ethics committee in Linköping. The approval number is ID 1278-18.

### Decision letter and Author response

Decision letter https://doi.org/10.7554/eLife.63284.sa1
Author response https://doi.org/10.7554/eLife.63284.sa2

## Additional files

### Supplementary files

- Source data 1. Exact P values.
- Transparent reporting form

## Data availability

Data generated in this study are presented in the manuscript and supporting files. A source file for exact P values is also included.

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
