## [Decision Letter]

**Acceptance summary:**

The manuscript by Bergo and colleagues seeks to test the effects of genetic and pharmacologic inhibition of isoprenylcysteine carboxylmethyltransferase (ICMT) on rescuing phenotypes of Hutchinson Gilford Progeria Syndrome (HGPS) in culture and in vivo in mice. Using progerin-knock-in mice, the authors show that a hypomorphic *Icmt* allele improves survival, restores vascular smooth muscle cell numbers in the aorta, and increases skeletal muscle fibre size. In addition, using a synthetic ICMT inhibitor referred to as C75, the author show its ability to rescue classical cellular and biochemical progeria hallmarks in HGPS patient fibroblasts and *Zmpste24*-deficient mouse fibroblasts, including premature senescence. The study presents an extension and confirmation of the authors' previous work showing that hypomorphic *Icmt* improves survival in *Zmpste24*-deficient mice, demonstrating a potential for ICMT inhibitors as new therapeutics against HGPS and other progeroid syndromes.

**Decision letter after peer review:**

Thank you for submitting your article "A small-molecule ICMT inhibitor delays senescence of Hutchinson-Gilford progeria syndrome cells" for consideration by *eLife*. Your article has been reviewed by two peer reviewers, and the evaluation has been overseen by a Reviewing Editor and Jessica Tyler as the Senior Editor. The following individual involved in review of your submission has agreed to reveal their identity: Susan Michaelis (Reviewer #3).

The reviewers have discussed the reviews with one another and the Reviewing Editor has drafted this decision to help you prepare a revised submission.

Summary:

The manuscript by Bergo and colleagues seeks to test the effects of genetic and pharmacologic inhibition of isoprenylcysteine carboxylmethyltransferase (ICMT) on rescuing phenotypes of Hutchinson Gilford Progeria Syndrome (HGPS) in culture and in vivo in mice. Using progerin-knock-in mice, the authors show that a hypomorphic *Icmt* allele improves survival, restores vascular smooth muscle cell numbers in the aorta, and increases skeletal muscle fibre size. In addition, using a synthetic ICMT inhibitor referred to as C75, the author show its ability to rescue classical cellular and biochemical progeria hallmarks in HGPS patient fibroblasts and *Zmpste24*-deficient mouse fibroblasts, including premature senescence. The study presents an extension and confirmation of the authors' previous work showing that hypomorphic *Icmt* improves survival in *Zmpste24*-deficient mice, demonstrating a potential for ICMT inhibitors as new therapeutics against HGPS and other progeroid syndromes. The reviewers agreed that the study provides a significant finding in establishing ICMT as a druggable target for HGPS and should be of broad interest to the community of researchers studying the biology of aging and HGPS. However, as detailed below, a number of concerns were also raised, relating in large part to insufficient clarity in the current version with respect to the authors' methods and the limitations of their approach and data.

Essential revisions:

1) This work is a continuation of their previous study in *Zmpste24* mice which slightly different from HGPS model used in the current study. The rescue on the HGPS mice is not surprising. However, the number of the mice presented in this study is far from sufficient in particular given that the generation of the *LAKI*^G609G/G609G^*Icmt*^hm/hm^ compound mutant mice seemed to be problematic, suggesting potential genetic variation due to modified gene. This caveat should be discussed in the text. Detailed answers to the following questions must be available.

a) What is the control body weight in Figure 1C? How the initial growth of the *LAKI*^G609G/G609G^*Icmt*^hm/hm^ compound mutant mice is affected during the first 2 months?

b) What exactly is the breeding problem resulting in difficulty in the production of *LAKI*^G609G/G609G^*Icmt*^hm/hm^ mice? If the *LAKI*^G609G/G609G^*Icmt*^hm/hm^ compound mutants are greatly rescued phenotypically, are they fertile?

c) Figure 1B, how many breeding has been made to obtain 3 *LAKI*^G609G/G609G^*Icmt*^hm/hm^ compound mutants? The number of the mice used in this analysis is too little. How long can the compound mutant mice survive? What is the average lifespan?

2) The lack of important controls in several figures presented, especially the data from WT cells, is a concern. Detailed answers to the following questions must be available.

a) The WT doubling should be included in Figure 2E, F and G, as an important control.

b) Figure 2I lack an important control to compare the growth of WT under C75 treatments with that in *LAKI*^G609G/G609G^*Icmt*^hm/hm^. If C75 inhibits prelamin A processing, it will produce unprocessed yet farnesylated prelamin A.

c) In Figure 3B. Authors should show what happen to WT cells in AKT phosphorylation in response to C75 treatment.

3) Why does Figure 2J, K showed inhibitory effects of FTI lonafarnib? If the FTI lonafarnib and C75 both inhibit the nuclear membrane localization of progerin, why does C75 promote HGPS proliferation whereas FTI lonafarnib does not? Figure 2K does not necessarily mean that C75 failed rescue HGPS cells in the absence of methylation. It can simply mean the inhibitory effect of FTI. What is the impact of C75 on WT cells in terms of nuclear shape and lamin A processing? Text and data presentation should be clarified to address these.

4) The treatment of WT and *Zmpste24* cells by C75 should have exactly the same effect given that RCE1 is involved in the first cleavage. Can authors explain why their data showed different response of WT and *Zmpste24*/HGPS cells? Text and data presentation should be clarified to address this point.

5) In Figure 3, authors showed that C75 treatment stabilized progerin therefore increasing the accumulation of progerin, specifically it nucleoplasm. It is very confusing as we know that it is the membrane progerin or prelamin A that gives rise to the increased DNA damage and senescence phenotype. Is the nuclear membrane-bound progerin/prelamin A or the increased nucleoplasmic prelamin A/progerin (farnesylate yet unmethylated) that results in reduced senescence? How? Text and data presentation should be clarified to address these points.

6) The only issue that detracts slightly from complete enthusiasm for this compelling study is the lack of thorough characterization of the new drug C75 as strictly a methylation inhibitor. Unanticipated effects of a drug on more than one target enzyme are not unprecedented, especially for lamin A processing enzymes. For instance, the HIV aspartyl protease inhibitor lopinavir unexpectedly is a zinc metalloprotease inhibitor for *Zmpste24* (Coffinier et al., 2007; PMID: 17652517). Likewise, a GGTI inhibitor was unexpectedly shown to block *Zmpste24* activity (Chang et al., 2012; PMID: 22448028). In the present study, C75 treatment causes accumulation of prelamin A (Figure 2B) and release of some RAS from the membrane fraction (Figure 2—figure supplement 2B), which are both expected outcomes of FTase inhibition. Is it possible that C75 could be inhibiting (albeit to a lesser degree than it inhibits ICMT) the farnesyltransferase complex? Perhaps the authors could look at another farnesylated substrate, such as HDJ-2, to show C75 has no effect on its mobility by SDS-PAGE? The mobility shift of HDJ-2 is often used as a test for farnesyltransferase inhibitors (FTIs), and would be a useful control for C75 treatment. Alternatively, or in addition, it could be helpful to test if C75 has any FTI activity in an in vitro assay. Text and data presentation should be clarified to address this point.

7) Importantly, the authors do make a significant effort to address the issue of C75 specificity to some extent, in that they show that proliferation of the *Zmpste24*^-/-^ cells is improved by C75 (Figure 2G) but proliferation of the *Zmpste24*^-/-^*Ictm*^∆/∆^ double mutant cells is unchanged upon C75 treatment (Figure 2H), suggesting genetically or pharmacologically blocking ICMT have same effect. Likewise, HGPS cells co-treated with C75 and FTI's (Figure 2K) abolishes the population doubling increase observed with C75 alone, (expected for FTIs hitting a step upstream of that inhibited by C75) strengthening the likelihood that C75 acts by inhibiting mainly ICMT in vivo. These two important figures lack error bars, suggesting the experiments should be repeated. Text and data presentation should be clarified to address this point.

8) If C75 were found to have some modest FTI activity in addition to inhibiting methylation, this would be an important piece of information to establish for this new drug. In any case, the main conclusion of this work – that C75 improves HGPS phenotypes – is clear and well supported. Text and data presentation should be clarified to address this point.

9) Mechanistically, there is little new information provided compared with their early study (Ibrahim et al., 2013). AKT phosphorylation was shown to be relevant to the rescue but no direct evidence to show blocking AKT phosphorylation attenuates C75 effect. Text and data presentation should be clarified to address this point.

10) No in vivo data were presented to show C75 could rescue the premature aging in HGPS mice. Text should be clarified to address this point.

---

## [Author Response]

Essential revisions:1) This work is a continuation of their previous study in Zmpste24 mice which slightly different from HGPS model used in the current study. The rescue on the HGPS mice is not surprising. However, the number of the mice presented in this study is far from sufficient in particular given that the generation of the LAKI^G609G/G609G^ Icmt^hm/hm^ compound mutant mice seemed to be problematic, suggesting potential genetic variation due to modified gene. This caveat should be discussed in the text. Detailed answers to the following questions must be available.

We agree with this important point (although survival extension was statistically significant with an n of 3 these numbers are lower than what we would have liked. Indeed, this was the reason we included the following text: “Although these data are statistically sound, they should be interpreted with caution as the mice were difficult to breed and we only obtained three double homozygotes.” It was also the reason we also analyzed the impact of *Icmt* deficiency on heterozygous *LAKI* mice. We are happy to provide more information as requested in your points below.

a) What is the control body weight in Figure 1C? How the initial growth of the LAKI^G609G/G609G^ Icmt^hm/hm^ compound mutant mice is affected during the first 2 months?

Because there was only one male mouse, we combined the body weight curves and showed change in body weight instead of actual body weight in Figure 1C. Author response image 1 shows graphs for the actual body weights of females (A) and males (B) from this graph. Specifically, the body weight of control females was 15–17 g at week 9 (63 days; peak) after which they continually lost weight (see Author response image 1). Body weight for male control mice was 18–23g at the peak (week 8–9) with subsequent weight loss. Importantly, after the control mice peaked in weight, the *LAKI^G609G/G609G^ Icmt^hm/hm^*mice remained at the peak weight and even continued to gain some weight.

**Author response image 1. sa2fig1:** 

b) What exactly is the breeding problem resulting in difficulty in the production of LAKI^G609G/G609G^ Icmt^hm/hm^ mice? If the LAKI^G609G/G609G^ Icmt^hm/hm^ compound mutants are greatly rescued phenotypically, are they fertile?

Thank you for this question. The problem is that Mendelian inheritance of the genes does not happen in the pups that are born and that fewer pups than normal are born. The answer to why this happens is unknown but it likely begins with the fact that the heterozygous *LAKI* mice are quite difficult to breed (smaller than normal litters). You are probably very well aware of these problems; but we would like to mention that from 20 years of experience with breeding genetically modified mouse strains, we have seen this phenomenon at least a dozen other times – i.e., that when we combine two or more alleles that on their own produce offspring with normal inheritance, no pups with double homozygosity are born. Sometimes this phenomenon goes away when the mice are bred onto a new genetic background, but in this case it didn’t. Because we have obtained only three double homozygotes thus far, these mice became too precious to be used for breeding, and we are afraid we can’t answer the question of whether they are fertile. We are continuing to breed these mouse strains, but will not be able to produce more for this particular submission. Since the current data is statistically significant, and we see similar effects with heterozygous *LAKI* mice (also significant), we hope that you will agree that this could be sufficient for the scope of this study, and that the robust rescue of the vascular phenotype (which is essential for the children with progeria) along with the drug data addresses new aspects of whether targeting ICMT would be effective in HGPS therapy.

c) Figure 1B, how many breeding has been made to obtain 3 LAKI^G609G/G609G^ Icmt^hm/hm^ compound mutants? The number of the mice used in this analysis is too little. How long can the compound mutant mice survive? What is the average lifespan?

Please see also the answers to point 1b for more details. For the *LAKI* allele, only heterozygotes (hets) are used in breeding and even they often produce smaller size litters than normal; for the *Icmt*^hm^ allele, both hets and homozygotes are used. The success rate should thus be that 1 in 4, 1 in 8, or 1 in 16 should be double homozygotes. We have bred to date 87 females and genotyped >300 pups. We can add that breeding the *Zmpste24*-knockout allele with *Icmt*^hm/hm^ was not this difficult. And again, because we only obtained three double homozygotes we killed them after all the control mice had died of progeria so that we could analyze the aortas and muscle fibers. Thus, we can’t answer the question of how long they survive. Please see Figure 1G for survival of the *LAKI* heterozygotes; these mice don’t show vascular or muscle phenotypes in the same way that homozygotes do, and thus we opted for survival experiments for the hets.

2) The lack of important controls in several figures presented, especially the data from WT cells, is a concern. Detailed answers to the following questions must be available.a) The WT doubling should be included in Figure 2E, F and G, as an important control.

We agree with this comment. In our original figure drafts the effect of the drug on WT human cells was included, but the curves became “messy”, crowded, and very difficult to read. Thus, we opted to show this data separately in the figure panel Figure 2I. As you can see, the drug only minimally affect proliferation of WT cells. We hope you will agree with this strategy. Thus, the control in terms of effects on WT cells is already included. Another control is shown in Figure 2H which is *Zmpste24*- and *Icmt*- double-deficient cells where we find that the ICMT inhibitor drug has no impact in cells already lacking the *Icmt* gene – an indication of drug specificity.

b) Figure 2I lack an important control to compare the growth of WT under C75 treatments with that in LAKI^G609G/G609G^ Icmt^hm/hm^. If C75 inhibits prelamin A processing, it will produce unprocessed yet farnesylated prelamin A.

We understand the comment but would like to mention that it is very difficult to compare human and mouse cell lines as they behave differently in terms of physical growth rate and population doubling time and we would therefore prefer to keep them separate as is. The reader can still compare the shape and slope of the curves side-by-side in the two panels. We hope you will agree with this argument.

c) In Figure 3B. Authors should show what happen to WT cells in AKT phosphorylation in response to C75 treatment.

Low levels of phospho-AKT is a reproducible finding in HGPS and *Zmpste24*-deficient cells – and these levels are increased and sometimes even restored to WT levels upon ICMT inhibition. WT cells already have high phospho-AKT levels and we had not included effects of C75 in these cells in our previous submission. Thus, in response to this important comment we analyzed levels of phosphorylated and total AKT in WT human cells incubated with C75. The western blots show that levels of phosphorylated and total AKT do not change in WT cells incubated with C75. The new western blot and quantification is added to Figure 3B along with comments in the Results and figure legend text. Thank you for this comment.

3) Why does Figure 2J, K showed inhibitory effects of FTI lonafarnib? If the FTI lonafarnib and C75 both inhibit the nuclear membrane localization of progerin, why does C75 promote HGPS proliferation whereas FTI lonafarnib does not? Figure 2K does not necessarily mean that C75 failed rescue HGPS cells in the absence of methylation. It can simply mean the inhibitory effect of FTI. What is the impact of C75 on WT cells in terms of nuclear shape and lamin A processing? Text and data presentation should be clarified to address these.

There are two main conclusions of Figure 2J and K: First, that the recently approved and only therapy for HGPS (i.e., FTI) is a drug with potent anti-proliferative properties (verified also by genetic strategies in previous studies) a result which we believe should increase the interest in targeting ICMT. Second, the cell-killing effect of FTIs is independent of ICMT-mediated methylation. This result makes sense as protein methylation can’t occur in the absence of farnesylation. FTase and ICMT process many protein substrates, in addition to prelamin A. The cell-killing effect of FTIs likely stems from reducing the farnesyl-dependent membrane targeting or function of several other proteins such as lamin B, CENP, RAS, or RHOA etc. It is fascinating to us that targeting ICMT which renders many proteins unmethylated can increase proliferation of HGPS cells—an indication that prelamin A methylation is an underlying cause and an indication that *CAAX*-protein methylation is dispensable for many cell functions.

Regarding the impact of C75 on nuclear shape of WT cells; since C75 (and *Icmt* knockout) does not influence nuclear shape in HGPS cells, it is highly unlikely that it would affect nuclear shape in WT cells where the levels of misshapen nuclei are low.

Regarding the question on the effects of C75 on lamin A processing in WT cells. We have previously shown that knockout of *Icmt* blocks prelamin A methylation and reduces the efficiency of the upstream ZMPSTE24-mediated cleavage resulting in partial prelamin A accumulation. To address your highly relevant question regarding the C75 drug, we performed western blots on C75-exposed WT cells. The data confirm the previous genetic analysis that ICMT inhibition causes prelamin A accumulation. The new data are added to Figure 2B, right panel, along with text additions in the Results and figure legends. Thank you for this suggestion.

4) The treatment of WT and Zmpste24 cells by C75 should have exactly the same effect given that RCE1 is involved in the first cleavage. Can authors explain why their data showed different response of WT and Zmpste24/HGPS cells? Text and data presentation should be clarified to address this point.

In wild-type cells, prelamin A is fully processed to mature lamin A and the cells grow and proliferate normally (i.e., mature lamin A is not farnesylated or methylated as the C-terminus including the farnesylmethylcysteine has been cleaved off); whereas In HGPS and *Zmpste24*-deficient cells, farnesylated and methylated progerin/prelamin A accumulates and causes senescence. Therefore, we don’t expect that C75 would produce the same effects in WT cells as in HGPS and *Zmpste24*-deficient cells. Our data suggest that progerin/prelamin A methylation contributes to the toxicity of these proteins and their ability to induce senescence; and we propose that blocking progerin/prelamin A methylation mislocalizes the proteins into the nucleoplasm and thereby reduces their ability to induce DNA damage, metabolic alterations, and senescence.

We hope this explanation is sufficient. Parts of this text has been added to the Results.

5) In Figure 3, authors showed that C75 treatment stabilized progerin therefore increasing the accumulation of progerin, specifically it nucleoplasm. It is very confusing as we know that it is the membrane progerin or prelamin A that gives rise to the increased DNA damage and senescence phenotype. Is the nuclear membrane-bound progerin/prelamin A or the increased nucleoplasmic prelamin A/progerin (farnesylate yet unmethylated) that results in reduced senescence? How? Text and data presentation should be clarified to address these points.

We agree with the assessment that nuclear membrane–bound progerin/prelamin A causes DNA damage and senescence. Therefore, we think it is fairly straightforward that if progerin becomes detached from the nuclear membrane—because it is no longer methylated, it can no longer cause senescence. We hope you will agree with this conclusion. This is now discussed in the main text starting on 113.

6) The only issue that detracts slightly from complete enthusiasm for this compelling study is the lack of thorough characterization of the new drug C75 as strictly a methylation inhibitor. Unanticipated effects of a drug on more than one target enzyme are not unprecedented, especially for lamin A processing enzymes. For instance, the HIV aspartyl protease inhibitor lopinavir unexpectedly is a zinc metalloprotease inhibitor for Zmpste24 (Coffinier et al., 2007; PMID: 17652517). Likewise, a GGTI inhibitor was unexpectedly shown to block Zmpste24 activity (Chang et al., 2012; PMID: 22448028). In the present study, C75 treatment causes accumulation of prelamin A (Figure 2B) and release of some RAS from the membrane fraction (Figure 2—figure supplement 2B), which are both expected outcomes of FTase inhibition. Is it possible that C75 could be inhibiting (albeit to a lesser degree than it inhibits ICMT) the farnesyltransferase complex? Perhaps the authors could look at another farnesylated substrate, such as HDJ-2, to show C75 has no effect on its mobility by SDS-PAGE? The mobility shift of HDJ-2 is often used as a test for farnesyltransferase inhibitors (FTIs), and would be a useful control for C75 treatment. Alternatively, or in addition, it could be helpful to test if C75 has any FTI activity in an in vitro assay. Text and data presentation should be clarified to address this point.

Thank you for these relevant comments and discussion. In response to this comment, we run HDJ-2 western in HGPS cells incubated with two concentrations of C75. We found C75 did not have any effect on HDJ-2 mobility. An important reason that we tend to believe the drug is specific for ICMT (and does not affect FTase) is that the drug produces essentially identical effects as *Icmt* deficiency. Indeed, knockout of *Icmt* – where FTase is fully functional – leads to RAS mislocalization and prelamin A accumulation (Bergo et al., JBC 2000; Bergo et al., 2002). Regardless, we agree with you that an FTase activity assay would be the best option to rule out non-specific effects on FTase, but we unfortunately do not have this assay set up. We hope you agree with us that the other data (including your excellent suggestion of HDJ-2 western blots) and arguments herein (and in point 7 below) are sufficient. Please see new figure panel (Figure supplement 2—figure supplement 2B) and brief text in the Results.

7) Importantly, the authors do make a significant effort to address the issue of C75 specificity to some extent, in that they show that proliferation of the Zmpste24^-/-^ cells is improved by C75 (Figure 2G) but proliferation of the Zmpste24^-/-^ Ictm^∆/∆^ double mutant cells is unchanged upon C75 treatment (Figure 2H), suggesting genetically or pharmacologically blocking ICMT have same effect. Likewise, HGPS cells co-treated with C75 and FTI's (Figure 2K) abolishes the population doubling increase observed with C75 alone, (expected for FTIs hitting a step upstream of that inhibited by C75) strengthening the likelihood that C75 acts by inhibiting mainly ICMT in vivo. These two important figures lack error bars, suggesting the experiments should be repeated. Text and data presentation should be clarified to address this point.

Thank you for this comment and for pointing out the missing error bars mistake. The experiment with *Zmpste24/Icmt* double knockout cells is shown in Figure 2H and actually did contain error bars (see the last data point enlarged): but the data is very reproducible and the technical replicates have very small differences. However, the error bars were definitely missing in Figure 2K and we have now added them. The experiments in all the cell population doubling experiments have been performed multiple times and are highly reproducible.

8) If C75 were found to have some modest FTI activity in addition to inhibiting methylation, this would be an important piece of information to establish for this new drug. In any case, the main conclusion of this work- that C75 improves HGPS phenotypes- is clear and well supported. Text and data presentation should be clarified to address this point.

Thank you for your comment. Please see answers to point 6 for a discussion on this topic and added text in the Results and a figure panel to Figure 2—figure supplement 2B.

9) Mechanistically, there is little new information provided compared with their early study (Ibrahim et al., 2013). AKT phosphorylation was shown to be relevant to the rescue but no direct evidence to show blocking AKT phosphorylation attenuates C75 effect. Text and data presentation should be clarified to address this point.

We agree with this comment. The goal of this study as outlined in the Introduction was to determine whether knockout of *Icmt* would improve phenotypes in an authentic progerin-expressing HGPS mouse model and to synthesize and pre-clinically validate a new ICMT inhibitor. The study does provide some new evidence that blocking methylation improves HGPS phenotypes, including the vascular and muscle fiber phenotypes which was not shown before; and new data on effects of C75 on cell respiration/metabolism/ROS levels/ER stress/DNA damage. To address your comment, we have added new text in the Discussion.

10) No in vivo data were presented to show C75 could rescue the premature aging in HGPS mice. Text should be clarified to address this point.

Yes, this is the main limitation of the study and we believe it is clearly shown and discussed in the revised manuscript.